# Endovascular Temporary Balloon Occlusion for Microsurgical Clipping of Posterior Circulation Aneurysms

**DOI:** 10.3390/brainsci10060334

**Published:** 2020-05-30

**Authors:** Jenny C. Kienzler, Michael Diepers, Serge Marbacher, Luca Remonda, Javier Fandino

**Affiliations:** 1Department of Neurosurgery, Kantonsspital Aarau, CH-5000 Aarau, Switzerland; jenny.kienzler@ksa.ch (J.C.K.); serge.Marbacher@ksa.ch (S.M.); 2Division of Neuroradiology, Department of Radiology, Kantonsspital Aarau, 5000 Aarau, Switzerland; michael.diepers@ksa.ch (M.D.); luca.remonda@ksa.ch (L.R.)

**Keywords:** aneurysm clipping, posterior circulation aneurysm, temporary balloon occlusion, intraoperative digital subtraction angiography, hybrid operating room

## Abstract

Based on the relationship between the posterior clinoid process and the basilar artery (BA) apex it may be difficult to obtain proximal control of the BA using temporary clips. Endovascular BA temporary balloon occlusion (TBO) can reduce aneurysm sac pressure, facilitate dissection/clipping, and finally lower the risk of intraoperative rupture. We present our experience with TBO during aneurysm clipping of posterior circulation aneurysms within the setting of a hybrid operating room (hOR). We report one case each of a basilar tip, posterior cerebral artery, and superior cerebellar artery aneurysm that underwent surgical occlusion under TBO within an hOR. Surgical exposure of the BA was achieved with a pterional approach and selective anterior and posterior clinoidectomy. Intraoperative digital subtraction angiography (iDSA) was performed prior, during, and after aneurysm occlusion. Two patients presented with subarachnoid hemorrhage and one patient presented with an unruptured aneurysm. The intraluminal balloon was inserted through the femoral artery and inflated in the BA after craniotomy to allow further dissection of the parent vessel and branches needed for the preparation of the aneurysm neck. No complications during balloon inflation and aneurysm dissection occurred. Intraoperative aneurysm rupture prior to clipping did not occur. The duration of TBO varied between 9 and 11 min. Small neck aneurysm remnants were present in two cases (BA and PCA). Two patients recovered well with a GOS 5 after surgery and one patient died due to complications unrelated to surgery. Intraoperative TBO within the hOR is a feasible and safe procedure with no additional morbidity when using a standardized protocol and setting. No relevant side effects or intraoperative complications were present in this series. In addition, iDSA in an hOR facilitates the evaluation of the surgical result and 3D reconstructions provide documentation of potential aneurysm remnants for future follow-up.

## 1. Introduction

Aneurysms of the posterior circulation, such as the basilar artery (BA), present a particular surgical challenge [1,2]. They represent 5–8% of all intracranial aneurysms and more than 50% of those in the posterior circulation [3,4]. Posterior circulation aneurysms are known to have a higher risk of rupture [5]. According to recently published scores such as PHASES [6] or UIATS [7], preventive endovascular or surgical methods can be performed in patients at risk, to minimize the chance of aneurysm rupture. The difficulties of microsurgical clipping are mainly caused by anatomical conditions and a demanding approach [8]. Surgical complexity varies according to size, shape, and localization of the aneurysm, degree of intraoperative brain swelling, and the microsurgical experience of the surgeon [9]. Moreover, standard clipping could fail due to insufficient proximal control and lead to incomplete occlusion or intraoperative aneurysm rupture [10]. Nevertheless, microsurgical clipping is still more accessible worldwide, especially in developing countries [11].

The ISAT (International Subarachnoid Aneurysm Trial) reported a higher rupture rate for basilar apex aneurysm in correlation with aneurysm size [12]. Increased morbidity and mortality, and worse clinical outcome was also reported after surgical clipping compared to endovascular coiling of a posterior circulation aneurysm [12]. These findings initiated the use of endovascular treatment for a basilar apex region aneurysm [13]. The fact that fewer neurosurgeons are performing microsurgical clipping of basilar apex aneurysm supports the trend for treating basilar artery aneurysms endovascularly rather than surgically [14]. The safety of endovascular occlusion of BA aneurysms has been proven, although long-term sustainability and efficacy remain unclear [13,14,15,16,17].

Up to 50% recanalization and regrowth of a coiled aneurysm has been reported [17,18,19,20]. The annual risk of bleeding in a partially coiled or recanalized aneurysm is reported to range from 2.1–15% [17,18,21,22,23]. This is relatively high and similar to rates for an unruptured aneurysm [5,24,25,26,27,28,29].

The exact location of the aneurysm is the key factor when deciding which surgical approach to take. The prevention of any injury to the brainstem and its perforators is crucial [30]. Different approaches to BA aneurysms have been described including the pterional approach introduced by G. Yasargil [31], the subtemporal approach pioneered by C. Drake [32], as well as lateral supraorbital [33], orbitozygomatic [34,35], modified presigmoid [36], transpetrosal [37] or transzygomatic transcavernous approaches, [13] and many others [38,39].

Various methods of temporary vessel occlusion or local blood flow interruption have been applied to facilitate a microsurgical approach to a large aneurysm in a narrow and deep location. Also, adenosine-induced cardiac arrest [40], hypothermic circulatory arrest [41], temporary clip placement [42], and temporary balloon occlusion [43] have been described. The relationship between the posterior clinoid process and the BA apex may limit the access for temporary clips [39]. An endovascular technique using balloon inflation in the parent vessel of the aneurysm can achieve proximal and distal control during surgery and, therefore, eliminate the need for temporary clipping [43]. Intraoperative temporary balloon occlusion (TBO) of the parent vessel might lower the risk of intraoperative rupture, reduce pressure in the aneurysm sac, and facilitate dissection and microsurgical clipping. The aim of this study is to describe the technical issues, setup, and experience of intraoperative TBO during surgical occlusion of complex posterior circulation aneurysms within the hybrid operating room (hOR). 

## 2. Materials and Methods

We report three cases of intracranial aneurysms of the posterior circulation that underwent clipping with the concurrent use of TBO in our department between 2013 and 2016. The first patient suffered subarachnoid hemorrhage (SAH) after the rupture of a basilar tip aneurysm (16 × 16 × 15 mm). Endovascular occlusion was not indicated due to the risk of occlusion of the posterior cerebral artery (PCA) and the superior cerebellar artery (SCA). The second case presented with a ruptured, partially thrombosed BA aneurysm (11 × 8 × 8 mm) with secondary wall hematoma and no SAH. As in the previous case, endovascular occlusion was considered not possible due to the risk of SCA occlusion caused by duplicate origin from the aneurysm fundus. The third patient had an incidental right proximal PCA (P1) aneurysm (4 × 5 × 5 mm). Endovascular treatment was scheduled, but 3D digital cerebral angiography (DSA) showed the SCA originating from the aneurysm sac and the treatment strategy was changed to surgery occlusion. Complex anatomical vascular findings were considered for the decision to choose a combined endovascular and microsurgical procedure within the (hOR) in these three cases (Table 1). 

The technical aspects of performed combined approaches in the hOR have been described by our group in an earlier publication [44]. The main unit consists of a 360° radiolucent carbon fiber table (Alphamaquet 1150, Maquet AG, Switzerland) that is coupled with the C-arm angiography system (Allura Xper FD20, Philips, Netherlands). A radiolucent head holder and pins are required for optimal acquisition of angiograms and intraoperative CT scans (Mayfield, Integra GmbH, Ratingen, Germany). A 7-Fr sheath is placed in the right or left femoral artery in preparation for intraoperative endovascular balloon occlusion and control DSA. All cases underwent an intraoperative DSA (iDSA) and CT (iCT) scan before they were transferred to the intensive care unit. 

### 2.1. Illustrative Cases

The surgical approach and endovascular techniques were similar in all three cases.

#### 2.1.1. Surgical Procedure

After the positioning of the patient′s head in a carbon clamp in the hOR, a right fronto-temporal craniotomy and selective extradural anterior clinoidectomy were performed. The proximal Sylvian fissure was opened, and the chiasmatic cistern incised, followed by dissection of the optic and oculomotor nerve, and carotid artery. Once the posterior clinoid process was exposed, a posterior clinoidectomy was completed with a 2 mm drill. The afterward visible BA, PCA′s, SCA′s, and aneurysm were inspected. After craniotomy, the first iDSA was performed by cannulation of the right femoral artery with a 7-Fr sheath and inserting a 5F diagnostic catheter in one of the vertebral arteries. The first iDSA showed the previously identified aneurysm and in Case 1 a progression of the dissecting basilar tip aneurysm with a new bleb. The diagnostic catheter was exchanged for a soft guiding catheter (Neuron 6F 058, Penumbra, Alameda CA, USA). An ASCENT^®^ 4 × 7 balloon (DePuy Synthes) and was then placed in the middle or distal segment of the BA and inflated under fluoroscopy to interrupt blood flow. The dual lumen design of the ASCENT balloon allows the distal flushing of the occluded vessel by saline. In the meantime, the BA or PCA aneurysm neck, which was significantly softened, as well as PCA and SCA branches were further dissected. The aneurysms were occluded in a microtechnical fashion with straight standard titanium 790-Yasargil-Clips (Aesculap, Tübingen, Germany) under visualization of both SCA branches. The balloon was deflated after 9, 10, and 11 min of TBO. An iDSA and intraoperative 3D-angiography in Case 1 showed complete occlusion of the BA aneurysm with patent PCA and SCA branches (Figure 1). The iDSA in Cases 2 and 3, revealed a small remnant at the aneurysm neck to preserve the SCA exit, no sign of aneurysm perfusion, and patent PCA and SCA branches (Figure 2 and Figure 3). The dura was sutured, the bone flap fixed, and the wound sutured using a standard multilayered technique. The iCT scan documented no hemorrhage or midline shift.

##### Case 1

History

This 53-year-old patient presented with a SAH after a sudden loss of consciousness at home. The patient was intubated upon admission with a Glasgow Coma Scale (GCS) score of 3. The CT scan showed a SAH caused by a ruptured basilar tip aneurysm (Fisher grade IV). A DSA was performed after the patient improved to a GCS of 10 following two days of conservative treatment and CSF drainage after ventriculostomy. A basilar tip aneurysm (16 × 16 × 15 mm) with the PCA and SCA bilaterally arising from the aneurysm base was identified. Indication for surgical occlusion was decided after interdisciplinary case discussion. The surgical procedure in the hOR had to be postponed for six days due to severe vasospasms in the posterior circulation. 

Postoperative Course

Further course on the ICU was unsuccessful. Consciousness persisted at a low level of GCS 5 due to severe vasospasm and the deterioration of cerebral perfusion on CT was observed within 48 h after surgery. Despite of endovascular spasmolysis with Nimodipine was performed no clinical improvement could be observed. During the next days, the general condition of the patient deteriorated due to pneumonia and respiratory failure leading to death eight days after surgery. 

##### Case 2

History

A 44-year-old patient was admitted to the emergency room with a thunderclap headache, neck pain, vomiting, and paresthesia in the right arm. The patient had a GCS score of 15 without meningism or any neurological deficits. Further investigations (CT, CTA, and DSA) excluded an SAH but showed a dissecting, partially thrombosed BA aneurysm (11 × 8 × 8 mm) with a secondary wall hematoma. Cerebral angiography revealed a duplicate origin of the SCA out of the aneurysm fundus. Treatment options were discussed in the interdisciplinary neurovascular board. In principal, endovascular occlusion by coiling or stent-assisted coiling was considered—with a high risk of SCA occlusion and secondary cerebellar ischemia. Therefore, the group decided to recommend microsurgical aneurysm clipping under endovascular TBO in the hOR. 

Postoperative Course 

The patient was hospitalized for one more week. During this time, the patient suffered an epileptic seizure. Apart from this, the patient could be rapidly mobilized with a GCS score of 15 and no new neurological deficits. 

##### Case 3

History

After the occurrence of vertigo, this 48-year-old patient underwent an MRI scan which revealed an incidental right PCA aneurysm. The patient was referred to our Institution, and the history and clinical examination excluded any episodes of headache, epileptic seizure, or neurological deficits. Further aneurysm imaging with DSA showed a saccular right BA aneurysm at the origin of the P1 segment of the PCA (4 × 5 × 5 mm). The pre-interventional 3D-DSA depicted the origin of the SCA directly arising from the aneurysm fundus, as well as a hypoplastic bilateral posterior communicating artery. After the case discussion with the neurovascular board, surgical occlusion with TBO in the hOR was recommended.

Postoperative Course

Postoperatively, the patient presented with a GCS score of 15, discrete ptosis, anisocoria, and double vision. A further CT scan confirmed otherwise regular findings (the ophthalmological symptoms were mainly caused by an impairment of oculomotor nerve function). The oculomotor nerve dysfunction had resolved by itself by the three-month follow up. 

## 3. Results

All cases underwent combined surgical and endovascular procedures in our hOR. After craniotomy and dissection of the parent vessel and aneurysm the intraluminal balloon was inserted through the femoral artery and inflated in the BA. In all three cases, intraoperative TBO was successfully performed without complications. No aneurysm rupture prior to clipping, or any other intra- or postoperative problems (necessary clip repositioning, parent vessel or branch occlusion, thromboembolic event, or re-bleeding) occurred. The mean duration of TBO was 10 min (Table 2). Upon inflation of the balloon, the intraluminal pressure releases and the aneurysm softens rapidly, which gives the surgeon more space and flexibility to explore the aneurysm and vessels branching out of the aneurysm base. Intraoperative DSA following clipping confirmed complete aneurysm occlusion with patent parent and branch vessels. In two cases, a small remnant at the aneurysm neck was visible in the intraoperative 3D angiography, which was necessary to preserve the branch origin. Two patients showed good postoperative recovery with GOS 5 and one patient died due to severe vasospasm and pneumonia.

## 4. Discussion

The findings of this technical note support the fact that TBO is a feasible, safe, and reliable method for the clipping of posterior aneurysms that are technically demanding and complex due to size or anatomy. In our institution, TBO with a combined endovascular and surgical approach is also used for giant and complex recurrent middle cerebral artery (MCA) or anterior communicating artery aneurysms. In cases with a ruptured MCA aneurysm and surrounding hematoma, endovascular TBO facilitated clipping following hematoma evacuation and prevented an intraoperative rupture of fragile high-risk aneurysms. 

In the case of intraoperative rupture, TBO could effectively control acute bleeding and increase the safety and accuracy of clip placement. In one case report with the intraoperative rupture of a paraclinoid aneurysm, TBO provided a salvage procedure for adequate hemostasis with additional intraluminal support to preserve the parent artery during clip placement [43].

In our opinion, microsurgery should still be viewed as a valuable option in the management of posterior circulation aneurysms. Various authors have reported good radiological and clinical outcomes for BA clipping with or without additional bypass [33,45,46,47,48,49,50,51]. Overall, the most common complications in this location are perforator and branch ischemia-related events and cranial nerve deficits, often involving oculomotor nerve palsies [47,52,53,54]. One case of transient oculomotor nerve palsy occurred in our series. 

The angiographic obliteration rate of posterior circulation aneurysm has been reported with a range from 91.9–98.1% [8,29,45,55]. However, other reports also cited 11.5% transient and 7.8% permanent neurological deteriorations [45].

Circumferential exposure of the aneurysm, including branches and perforators, is necessary prior to a safe and efficacious clip application. Dissection and visualization of where they exit the aneurysm can be very demanding. Various methods have been described to support the surgeon during the clip application. Additional endovascular assistance can help prospective vascular neurosurgeons to become more confident and proficient in these cases. Temporary parent vessel occlusion seems to be a safe procedure. Interestingly, a study with a mean follow-up of 53 months showed that a temporary artery occlusion time (mean 16.1 min) had no effect on overall long-term clinical outcomes [56].

The “gold standard” for proximal vessel control is a temporary clip application, which is not always feasible, especially in areas with limited access [57]. Proximal parent vessel ligation [58] can be considered for treating giant aneurysms. Transient asystole with adenosine [40,59,60], deep hypothermic circulatory arrest [61], or rapid ventricular pacing [62] are other techniques also described in the literature but have higher risk profiles for side effects such as atrial or ventricular fibrillation, arrhythmias, and prolonged hypotension [62]. In addition, the risk of stroke can increase after a circulatory arrest, and the resulting need for a multidisciplinary team of surgeons and technologists is logistically challenging and expensive [63]. Perioperative morbidity and mortality for circulatory arrest has been described as ranging from 8.3–17% [41,61]. The risk of side effects of these different techniques has to be weighed against the significant chance of intraoperative aneurysm rupture, incomplete clipping, or unintended branch occlusion due to poor visualization. 

In comparison, TBO presents a simple, fast, and inexpensive technique with no need for special anesthesiological monitoring or training and can be performed at any time in every center with endovascular expertise. Although temporary clipping will remain the routinely used technique, TBO may be more accurate for complex and large aneurysms, especially in the posterior circulation and to prevent premature rupture. In posterior circulation aneurysm clipping, proximal control with temporary clipping is often not possible due to the complex anatomy, skull base proximity, and location near the brain stem and cranial nerves [64]. The temporary clip itself may hinder the placement of the permanent one due to the limited surgical corridor [65]. In these cases, TBO can provide a reliable alternative. Proximal control with TBO can be achieved before craniotomy with minimal obstruction of the surgical field and less brain retraction. 

Possible side effects of TBO and temporary clipping include wall injury of the parent vessel or thromboembolic events causing postoperative ischemic deficits. MacDonald et al. compared the degree of acute endothelial injury after temporary vessel occlusion with external clipping and endovascular balloon occlusion in a pig model [66]. The results revealed that vessel injury worsened with time and was more prevalent adjacent to the clip; as compared to the widespread pattern with TBO [66]. There is the concern of a higher risk of ischemic complications with balloon occlusion in perforator-rich vessels like MCA and BA, but neither MCA nor BA TBO interventions at our institution led to perforator infarctions. 

The TBO technique was first described in 1986 by Kinjo et al. [67], by Shucart et al. in 1990 [68], and in many other case series since then. More recently, another group has reported on the use of TBO in the hybrid OR [69]. Table 3 provides an overview of the literature to-date. 

Most series included large or giant paraclinoid ICA aneurysms occluded with clip ligation after balloon catheter placement in the ICA [69,70,71]. Intra-luminal pressure was decreased through the additional placement of a temporary clip distal to the aneurysm on the posterior communicating artery to reduce collateral blood flow, as well as the application of the retrograde decompression-suction method in the ICA [69,71,72]. In these series, endovascular TBO eliminated the need for cervical ICA dissection [68,69]. A review of the literature found a total of 188 aneurysms clipped with TBO. The largest series, published by Fulkerson et al [73]. included 63 ophthalmic artery aneurysms. The description of TBO in posterior circulation aneurysms, however, is less common with a total of only 20 cases. Bailes et al. published the first series of TBO use in multiple basilar artery aneurysms [74]. Apart from the recent study, eight other series have used TBO only for successful clipping [64,65,68,70,74,75,76,77]. Balloon placement in the aneurysm orifice or neck has only been described in two case series [65,77] and was otherwise performed in the proximal parent vessel. TBO duration ranged from 1.5–3 min for each balloon inflation and a total maximum of 50 min [43,63,64,65,68,71,75,77,78,79,80,81].

The overall reported complication rate for TBO is very low at 1.7–3.7% [72]. TBO procedure-related thromboembolic events occurred in five patients (2.6%). This risk is increased in cases with pronounced vessel wall sclerosis or prolonged temporary occlusion. One intraoperative balloon rupture and balloon exchange led to a thrombus in the M1 segment, with subsequent intraoperative embolectomy and postoperative transient hemiparesis. Symptoms such as dysphasia and hemiparesis were transient in all other cases except one major MCA infarct, which lead to the death of the patient [71,80,81,82]. Further TBO-related complications included: ICA intima dissection with ICA occlusion at the neck requiring medical treatment only (recovery was complete after four days) [81], as well as increases in vasospasms due to mechanical wall stimulation with transient hemiparesis [77].

Thromboembolic events may be reduced by limiting TBO duration and using double-lumen balloon systems that allow for maintaining continuous heparinized saline catheter flush. In thrombosed aneurysms or patients with severe atherosclerosis, the risk for thromboembolism might be increased. No complications occurred in our series that had a mean TBO time of 10 min. Several series suggested multiple short inflation times of 1.5–5 min to reduce thromboembolic event rate [63,65,78]. Some studies used a preoperative TBO test [83] to investigate the capacity of collateral support. In view of the short TBO time in our series, it is questionable if this is needed. The advantage of performing a TBO procedure in the hybrid OR is that control angiography is possible immediately after clip placement. We performed 2D and 3D intraoperative angiography in the hybrid OR and confirmed aneurysm occlusion in all cases. This standardized protocol can achieve better outcomes [84,85].

The main limitation of our study is the small sample size, as we chose to report on posterior circulation aneurysms only. 

## 5. Conclusions

Intraoperative endovascular TBO is a feasible, safe, and valuable procedure for surgical treatment of complex posterior circulation aneurysm undergoing clipping. In addition, intraoperative DSA and 3D-DSA in the hOR was confirmed as a valuable tool for the evaluation of aneurysm occlusion and possible aneurysm remnants.

## Figures and Tables

**Figure 1 brainsci-10-00334-f001:**
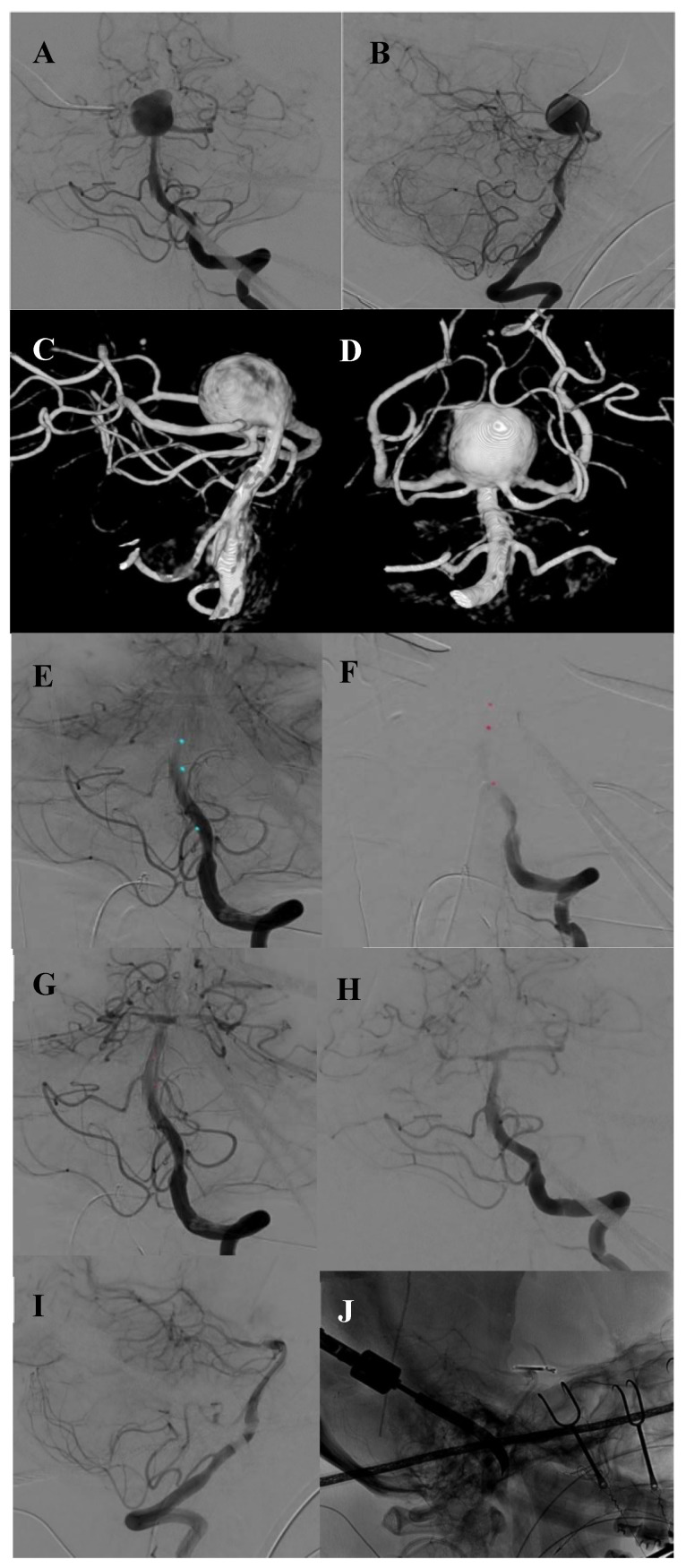
Case 1. Preoperative ap and lateral DSA of the basilar tip aneurysm (**A**,**B**). Preoperative 3D angiography of the basilar aneurysm presenting the bilateral origin of the PCA and SCA from the aneurysm base (**C**,**D**). Intraoperative angiography showing the endovascular placement of the balloon and occlusion of the basilar artery (**E**,**F**). Intraoperative DSA after clipping and closure of the balloon showing complete occlusion of the aneurysm in ap and lateral view with all branches open (**G**–**I**). Intraoperative picture of the opened skull, placed fish hooks, spatula and clip (**J**).

**Figure 2 brainsci-10-00334-f002:**
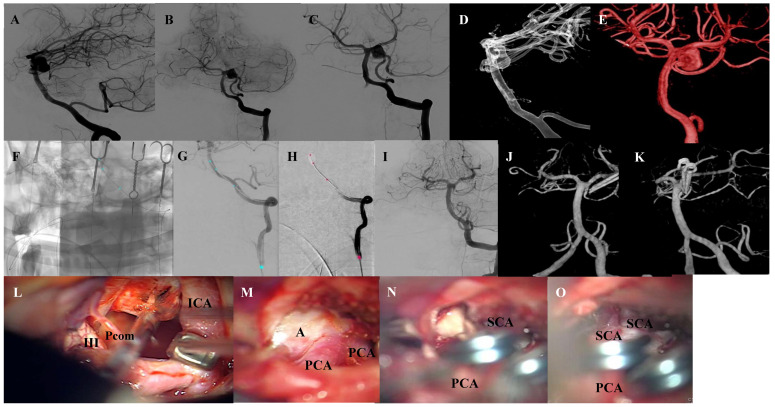
Case 2. Preoperative DSA ap and lateral projections of the left side dissecting basilar artery aneurysm (**A**–**C**). Preoperative 3D-DSA of the basilar aneurysm, presenting the ampullary exit of the left double-laid SCA from the aneurysm fundus (**D**,**E**). Intraoperative DSA showing the endovascular placement of the balloon and occlusion of the basilar artery through balloon inflation (**F**,**G**). Intraoperative angiography after clipping and deflation of the balloon demonstrating occlusion of the aneurysm in the ap and lateral view (**H**,**I**). Intraoperative 3D angiography in the ap and lateral view, showing a small remnant at the neck of the aneurysm to preserve the SCA exit (**J**). Microsurgical view showing the aneurysm approach with mobilization of the ICA with hook retractor (**K**), BA aneurysm (**L**), and situs after clipping with patency of all branches and parent vessel (**M**–**O**). Abbreviations: DSA = digital subtraction angiography, SCA = superior cerebellar artery, ICA= internal carotid artery, BA= basilar artery, III = oculomotor nerve, PCA = posterior cerebral artery, A = aneurysm, Pcom = posterior communicating artery.

**Figure 3 brainsci-10-00334-f003:**
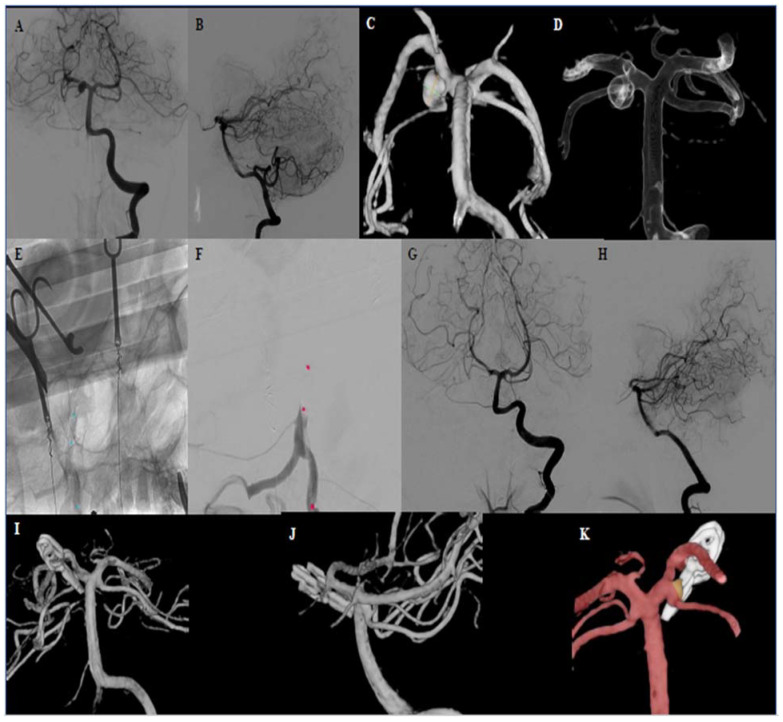
Case 3. Preoperative DSA ap and lateral projections of the proximal right side PCA aneurysm (P1 branch) (**A**,**B**). Preoperative 3D-DSA of the PCA aneurysm, showing the right SCA leaving the P1 segment from the aneurysm sidewall (**C**,**D**). Intraoperative DSA presenting the placement of the balloon and occlusion of the BA after balloon inflation (**E**,**F**). Intraoperative angiography after clipping and balloon removal demonstrating occlusion of the aneurysm and patency of the BA and PCA (**G–I**). Intraoperative 3D-DSA showing a small remnant at the aneurysm neck to preserve the patency of the SCA (**J**,**K**). Abbreviations: DSA = digital subtraction angiography, PCA = posterior cerebral artery, SCA = superior cerebellar artery.

**Table 1 brainsci-10-00334-t001:** Demographic characteristics, clinical findings, aneurysm description, and outcome parameters.

Case	Age (Years)	Gender	Aneurysm Location and Anatomical Variation	Aneurysm Size (mm)	Previous Treatment	Clinical Presentation	Fisher Grade	H and H/ WFNS Grade	GCS	mRS	GOS
**1**	53	Female	Basilar tip aneurysm Bilateral PCA and SCA are leaving from the aneurysm base	16 × 16 × 15	no	Dizziness attack and syncope, SAH	4	5/5	3	6	1
**2**	44	Male	Partially thrombosed distal left side dissecting basilar artery aneurysm with secondary wall hematoma Ampullary exit of the left double-laid SCA from the aneurysm fundus	11 × 8 × 8	no	Thunderclap headache	N/A	N/A	15	0	5
**3**	48	Female	Proximal right side PCA aneurysm (P1 branch) Right SCA exits the P1 segment from the aneurysms side wall	4 × 5 × 5	no	Incidental finding Diagnostic in the context of a vestibular syndrome	N/A	N/A	15	0	5

PCA = posterior cerebral artery, SCA = superior cerebellar artery, SAH = subarachnoid hemorrhage, N/A = not available, H and H = Hunt and Hess, GCS = Glasgow Comma Scale, mRS = modified Rankin Scale, GOS = Glasgow Outcome Scale.

**Table 2 brainsci-10-00334-t002:** Details of the intraoperative balloon occlusion procedure, clipping, and iDSA findings.

Case	Duration of TBO (Min)	Number of Clips	Intraoperative DSA Findings	Balloon Catheter Used	Complications
**1**	9	1	Complete occlusion of the aneurysm All branches open	ASCENT^®^ 4 × 7 balloon (DePuy Synthes)	None
**2**	10	2	Small remnant at the neck to preserve the SCA exit No aneurysm perfusion All branches open	ASCENT^®^ 4 × 7 balloon (DePuy Synthes)	None
**3**	11	3	Small remnant at the neck to preserve the SCA exit No aneurysm perfusion All branches open	ASCENT^®^ 4 × 7 balloon (DePuy Synthes)	None

iDSA = intraoperative digital subtraction angiography, TBO = temporary balloon occlusion, SCA = superior cerebellar artery.

**Table 3 brainsci-10-00334-t003:** Overview of all aneurysm cases including clipping with TBO in the literature. Information includes aneurysm characteristics, TBO duration, additional techniques, aneurysm occlusion status and complications. All posterior circulation aneurysms and complications related to TBO are listed in bold type.

Author	Aneurysm Location	TBO Occlusion Time	Additional Techniques	Complete Occlusion	Complications
Kinjo T. et al. [41] 1986	**Giant BA aneurysm**	N/A	Sendai cocktail Temporary clips on basilar artery and bilateral PCA	N/A	None
Shucart W.A. et al. [71] 1990	**BA aneurysm** 3 × large paraclinoid ICA aneurysms	3–18 min (mean 12.5 min.)	None	3 × optimal clip placement 1 × clip repositioning	Transient oculomotor palsy Oculomotor and abducens palsy Hemiparesis due to vasospasm Aphasia due frontal infarct (Moya Moya disease)
Tamaki N. et al. [76] 1991	Four large and giant carotid-ophthalmic artery aneurysm	N/A	Temporary ICA clipping distal to aneurysm ICA blood aspiration through second catheter	Successful clipping	Transient oculomotor palsy Hemorrhagic infarction Hydrocephalus (*n* = 2)
Scott J.A. et al. [68] 1991	Large ophthalmic artery aneurysm	1.5 min. per balloon inflation	Temporary ICA clipping distal to aneurysm Suction through distal lumen of occlusion balloon catheter	Successful obliteration	None
Bailes J.E. et al. [4] 1992	**4 BA aneurysms**	N/A	None	Complete obliteration	Intraoperative aneurysm rupture Transient oculomotor nerve palsy in 3 patients
Albert F.K. et al. [2] 1993	2 proximal paraclinoid aneurysm	N/A	Suction-decompression Temporary clip distal ICA	N/A	None
Mizoi K. et al. [51] 1993	9 paraclinoid ICA aneurysms	13–50 min. (mean 26.2 min.)	Retrograde suction-decompression Repeated TBO	3 × Clip repositioning (due to narrowing of parent artery) Successful obliteration in all cases	Intraoperative TBO balloon rupture → surgery continued with new balloon →Embolus developed in M1 Segment → Embolectomy through M1 incisio, MCA occlusion for 20 min → aneurysm clipping followed → patient with transient hemiparesis for 6 h
Mizoi K. et al. [52] 1994	**5 BA aneurysms** (including 1 large & 1 giant)	15–30 min. (mean 22 min.)	None	3 cases successful obliteration 2 × clip repositioning: Finally one case with remnant at the neck, one complete occlusion	Transient abducens nerve palsy Transient hemiparesis
Fahlbusch R. et al. [19] 1997	3 giant paraclinoid ICA aneurysms	2–6 min.	Retrograde suction-decompression	Clip repositioning in all 3 cases (stenosis of parent vessel or incomplete occlusion)	Vasospasm and temporoparietal infarction (hemiparesis and transient dysphasia, deterioration of vision) 1 thromboembolic complication: sensimotor dysphasia, infarct in temporo-parietal region
Hacein-Bey L. et al. [28] 1998	1 large & 1 giant ICA ophthalmic aneurysm	N/A	None	Complete obliteration	None
Arnautovic K.I. et al. [3] 1998	8 giant and 8 large paraclinoid aneurysm	mean 10.7 min.	Distal temporary clip Suction-decompression	In 1 patient clipping was not possible due to calcification of aneurysm neck and wall, no aneurysm collapse achieved → Coiling Successful clipping in 15 cases	Transient oculomotor nerve palsy 2 temporal lobe infarction with new neurological deficit Hydrocephalus and subdural hygroma Ipsilateral ICA intimal dissection and occlusion at balloon side Transient dysphasia (thromboembolic) Cerebral abscess Transient cerebral edema One death due to intraventricular hemorrhage
Ng P-Y. et al. [60] 2000	24 paraclinoid ICA segment aneurysms (13 large & 11 giant)	2–27 min. (mean 13 min.)	Suction-decompression in 16 cases	Clip adjustment in 7 cases (residual filling of aneurysm and 4 ICA compromise) Complete obliteration in 20 cases, greater than 90% occlusion in 22 cases	1 major MCA infarct related to catheter thromboembolism vasospasm in 3 cases with delayed ischemia 2 deaths (one from fatal MCA infarct)
Thorell W. et al. [80] 2004	6 paraclinoid giant or complex ICA aneurysms	Balloon inflation time less than 3 min. in all cases	None	In 4 patients, navigation of the balloon into intracranial circulation due to carotid tortuosity was not possible All aneurysms were complete occluded	Lumbar drain due to subgaleal CSF collection Lacunar infarction ipsilateral 6 weeks after surgery with mild transient hemiparesis
Steiger H.J. et al. [74] 2005	2 giant carotid opthalmic aneurysms	16 + 24 min. (mean 20 min.)	None	Case 1: Complete occlusion of the intracranial aneurysm part at surgery and 5 months later, endovascular occlusion of infraclinoid aspect of aneurysm Case 2: Small remnant at the neck for patency of parent vessel	Acceleration of vasospasms 15 h after surgery and transient hemiparesis in one case, resolved with hypertensive therapy
Ricci G. et al. [65] 2005	5 giant paraclinoidal ICA aneurysms **1 giant vertebrobasilar junction aneurysm**	N/A	None	Aneurysm obliteration achieved in all cases	None
Parkinson R.J. et al. [63] 2006	Giant paraclinoid ICA aneurysm		Suction-decompression Temporary clip distal to aneurysm	Complete occlusion and reconstruction of parent vessel	None
Petralia B. et al. [64] 2006	10 giant paraclinoid ICA aneurysm **3 giant vertebrobasilar aneurysm**	15–20 min.	None	All aneurysm excluded with parent vessel patency One balloon rupture One patient underwent coiling at recurrence	None
Hoh D.J. et al. [32] 2008	Large paraclinoid aneurysm	11 min.	Retrograde suction-decompression Temporary clip distal to aneurysm	Aneurysm completely occluded	None
Fulkerson D.H. et al. [23] 2008	63 opthalmic artery aneurysm (26 small, 23 large, 14 giant)	N/A	Temporary clip distal to aneurysm Suction-decompression	N/A	Stroke Hematoma requiring treatment Intraoperative aneurysm rupture New visual deficit 5 deaths
Elhammady M.S. et al. [16] 2009	Large paraclinoid aneurysm	20 min.	Prior attempt for with temporary clipping, thereafter aneurysm rupture and switch to TBO as salvage technique	Complete obliteration	Hemiparesis due to stroke in the inferior division of MCA Vasospasm requiring intra-arterial nicardipine infusion
Skrap M. et al. [72] 2010	11 giant paraclinoid aneurysm **4 vertebrobasilar aneurysm**	15–20 min. (Average 17 min.) Max 5 min. per balloon inflation	One balloon below the aneurysm neck and another in the PCA to stop retrograde flow from ICA	One balloon rupture intraop One clip repositioning in BA Complete occlusion in all cases	None
Dehdashti A.R [13] 2015	Large ophthalmic artery aneurysm	N/A	Temporary clip distal to aneurysm (Pcomm and ICA)	Complete occlusion and patent ophthalmic artery	None
Matano F. et al. [49] 2017	2 large ICPC aneurysms	N/A	STA-MCA Bypass prior to clipping Temporary clips on M1, A1 and Pcomm Retrograde suction decompression	Complete occlusion of aneurysms and patent parent vessel	None
Capo G. et al. [11] 2018	**Giant VA aneurysm**	N/A	Temporary clip on PICA	Complete aneurysm occlusion	Transient dysphonia Transient dysphagia Facial nerve palsy PICA occlusion
Xu F. et al. [69] 2018	Large paraclinoid aneurysm	N/A	Temporary clip on Pcomm Retrograde suction-decompression	Complete obliteration of aneurysm	None

TBO = temporary balloon occlusion, PCA =posterior cerebral artery, N/A = not available, BA = basilar artery, min = minutes, TBO = temporary balloon occlusion, ICA = internal carotid artery, MCA = middle cerebral artery, CSF = cerebro-spinal fluid, ICPC = internal carotid posterior communicating artery aneurysms, Pcomm = posterior communicating artery aneurysm, VA = vertebral artery, PICA = posterior inferior cerebellar artery.

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
