# Peer review of "Endovascular Temporary Balloon Occlusion for Microsurgical Clipping of Posterior Circulation Aneurysms"

_brainsci, 2020, doi:10.3390/brainsci10060334_

Round 1

Reviewer 1 Report

In this interesting paper the Authors have reported their experience with TBO in vascular neurosurgery.

These are my suggestions:

  • the introduction paragraph is too redundant. I suggest to review it.
  • where is the novelty of this paper? Please specify better this. The Authors have demonstrated that there are so many articles regarding this topic.

Author Response

We thank the reviewer 1 for his comments. Redundant sentence within the introduction were deleted. The novelty of the paper relies on the description of this intraoperative technique within the hybrid operation room. Although this technique has been described before, the present study highlighted the technical issues and setup of this technique within an operation room allowing microsurgical and endovascular techniques within the same setting.

Please see attached revised manuscript.

Reviewer 2 Report

I read with interest the paper of Kinzler et al. The authors described the use of endovascular Temporary Balloon Occlusion (TBO) before microsurgical clipping of posterior circulation aneurysms in 3 patients. The duration of TBO varied between 9 and 11 minutes and no patient had permanent neurological complications. Cases description was accurate and included an overview of all similar cases reported in the literature. Moreover, the manuscript was enhanced with appropriate and pertinent iconography. I have just a few minor suggestions for the authors.

Page 3 line 97: Please, specify what is hOR.

Page 8 lines 170-171: Please report for how many days the surgery has been postponed.

Page 8 line 199: It is curious that carbamazepine was prescribed for vertigo. Please check this information.

Page 9 line 231: Change “temporary balloon occlusion” with TBO.

Page 14 line 333: I think that a number of 85 references is too high. Please, consider to decrease it.

Author Response

We thank the reviewer 2 for suggestions. Following correction were inserted:

Page 3 line 97: Please, specify what is hOR. > 

The abbreviation of hOR was insert in the introduction.

Page 8 lines 170-171: Please report for how many days the surgery has been postponed.

"six days" was inserted

Page 8 line 199: It is curious that carbamazepine was prescribed for vertigo. Please check this information.

The sentence "Carbamazepine was described for vertigo" is not relevant for the case presentation and was deleted.

Page 9 line 231: Change “temporary balloon occlusion” with TBO.

TBO was inserted.

Page 14 line 333: I think that a number of 85 references is too high. Please, consider to decrease it.

The number of references was significantly reduced after shortening the introduction.